

**Hydrology and beyond: The scientific work of August Colding revisited**
Dan Rosbjerg, Department of Environmental Engineering, Technical University of Denmark,
Bygningstorvet, Building 115, DK 2800 Kongens Lyngby
Email: daro@env.dtu.dk
ORCID: 0000-0003-2204-8649
**Abstract.** August Colding was one of the three pioneers, who in the mid-1800s almost
simultaneously and independently formulated the first law of thermodynamics, the two others being
Robert Mayer and James Joule. This first, significant achievement was followed by a sequence of
other ground-breaking discoveries within a broad range of disciplines: magnetism, steam power, gas
production, hydraulics, soil physics, hydrology, heating and ventilation, meteorology and
oceanography. Moreover, he gave a significant contribution to the understanding of the spread of
cholera. In hydrology, he used evaporation experiments to obtain water balances. Independently, he
formulated Darcy's law, and, as the first one, he calculated the water table between drainpipes and
the piezometric surface in confined aquifers. His main occupation, however, was chief engineer in
Copenhagen, where he modernised the city by introducing groundwater-based water supply and
building a waterworks delivering pressured, clean water into the houses, a gasworks and gas-based
street lightening, and a citywide sewage system. Colding has not been recognised internationally as
he might deserve, probably because most of his publications were written in Danish. Even in
Denmark, he seems today almost forgotten. This paper highlights his most important scientific
contributions, in particular his achievements in hydrology, hydraulics, meteorology and
oceanography.

**1. Introduction**
Ludvig August Colding (1815-88) grew up on a farm close to Copenhagen but showed no interest in
becoming a farmer. Instead, he was trained as a cabinet-maker after advice from Hans Christian
Ørsted (the world-renowned physicist who discovered electro-magnetism) to whom his father was
acquainted. This first training raised his interest for engineering, and, having passed the entrance
examination, he started at the Polytechnic School, where Ørsted was director. During his study, he
assisted Ørsted by measuring the heat released from compression of water. He graduated in 1841,
and after a couple of years with miscellaneous teaching activities, he was employed by the city of
Copenhagen, first as road/bridge inspector and from 1845 as water inspector. In 1857, he was
promoted to become the first chief engineer in Copenhagen, a position he held until retirement in
1883. A photo of August Colding is seen in Fig. 1.

**2. The first law of thermodynamics**
During Colding's studies and his work as Ørsted's protégé, he became strongly interested in the
nature of forces (motive force, heat, electricity and chemical forces), and their possible
disappearance. In 1843, he submitted a treatise concerning forces to the Royal Danish Academy of



Sciences, but not printed before 1856 (Colding 1843/1856). He knew d'Alembert's principle for the
equilibrium of lost forces. However, Colding's belief was that when and wherever a force seems to
vanish in performing certain mechanical, chemical, or other work, the force then merely would
undergo a transformation and reappear in a new form, but of the original amount (Caneva, 1997).
Thus, Colding claimed the imperishability of forces (i.e. the energy).
To prove his statement, he performed a series of experiments, where a sled loaded with different
cannonballs was dragged on rails of different metals. By measuring the heat expansion of the rails,
he concluded that when we employ a motive force to overcome the resistance, which a body
experiences in sliding over other bodies of quite different nature, the heat evolved from the friction
is strictly proportional to the work expended (Dahl, 1972). Following experiments using an improved
experimental setup, see Fig. 2, enabled him to estimate the mechanical equivalent of heat (Colding,
1851a). Unfortunately, Ørsted found it difficult to follow Colding's idea of imperishability of forces,
which significantly delayed the publication of the treatise. Despite this delay, Colding claimed
priority to the discovery, as did Mayer and Joule (Kragh, 2009). The general perception of today is
that all three should be considered equal. In the above, the term "force" has been kept in the same
way as Colding did. In parallel, he also used the term "activity" of forces. In the first half of the
1800s, the term "energy" was not yet introduced as the work of a force.
In 1851, Colding demonstrated the generality of his theorem by also taking fluids and gasses into
account (Colding, 1851b), and he showed that the theorem could be used to improve the efficiency
of steam engines (Colding 1851c; 1853). He expanded further by a detailed investigation of the
forces involved in magnetisation of soft iron (Colding, 1851d). Much later the essence of his work on
the first law of thermodynamics was translated into English and French (Colding, 1864a; 1864b;
1871a).

**3. Responsible engineer for water supply, gas lightening and sewerage in Copenhagen**
At that time, when Colding was employed by the city of Copenhagen, the water supply was
insufficient and unhygienic. It was based partly on polluted wells in the city and partly on water from
small surrounding lakes that was led into the city through leaky wooden pipes. Moreover, there was
no sewerage, and the smell was terrible. In 1849, the city decided to remedy the situation and
launched international competitions on, respectively, water supply, gas lightning and sewerage
projects. Colding won the water supply competition with an innovative project suggesting that the
surface water from a lake west of Copenhagen should be supplemented with groundwater from
surrounding artesian wells. Sand filtering was introduced, and a water works powered by steam
engines built along with a clean water consumption-equalizing reservoir close to the city.
Before the three projects were finally decided, a cholera epidemic hit Copenhagen in 1853. Almost
5000 people died. Together with a younger colleague, Julius Thomsen (a later famous chemist),
Colding immediately investigated the causes for the spread of the disease, and they found that the
spread was strongly correlated to the soil water quality (pollution) and the population density in
different city sections. The path-breaking work was published already the same year (Colding &
Thomsen, 1853). A year later, John Snow in London gave the final proof for the cholera to be water
borne.
Colding was appointed managing engineer for all the above three projects. While the water supply
and the gas lightning projects were smoothly approved and initiated, the government heavily
delayed the sewerage project despite the tragic cholera epidemic by requiring extensive


85 modifications. Colding had to revise the winning far-sighted project that was based on separation of
86 rainwater and sewage water, a project that was 100 years ahead of time. Instead, a combined sewer
87 was decided without allowance for water closets, however still a significant improvement. The new
88 water supply and the gas lightning projects were inaugurated in 1859, while the sewerage project
89 was accomplished during the 1860s.

90

91 **4. The laws for water flow in filled and partly filled conduits**

92 At the time when Colding should design the sewerage in Copenhagen, there was no established
93 practise for dimensioning of conduits, form of the conduits (prismatic, circular of oval-shaped), and
94 their material. In particular, the capacity of a given pipe (the flow rate) was under discussion. A
95 sewerage commission in London had performed some full-scale experiments, where they found that
96 Eytelwein's formula was not applicable (a pre-runner of the formula today known as the Darcy-
97 Weisbach equation for steady, uniform flow in pipes and canals). Colding was not convinced and
98 performed a thorough theoretical analysis coming to the opposite conclusion, i.e. that Eytelwein's
99 formula (Eytelwein, 1842) in fact was valid and that the seemingly divergence was because the pipe
100 system had to be filled up before the formula was applicable.

101 To obtain a further solid foundation for the project it was decided that Colding should perform large-
102 scale experiments in Copenhagen with salt-glazed pipes. Two series of circular pipe experiments
103 were carried out with dimensions of, respectively, 4 inches and 12 inches, see Fig. 3. The length of
104 the pipes was 297 feet (almost 100 m). The experiments were carried out both with filled and partly
105 filled pipes. The hydraulic head could be monitored at 12 stations along the pipe, and it was made
106 possible to add water to a partly filled pipe at intakes along the conduits. Colding found that the
107 added water only had a local effect, implying the Eytelwein formula to be applicable in general. The
108 experimental measurements were supplemented by thorough, theoretical considerations including
109 calculation of the friction factor (Colding, 1857).

110

111 **5. Evaporation, percolation and water balance**

112 To evaluate the available amount of water for the new water supply to Copenhagen, Colding
113 initiated precipitation measurement near four lakes around Copenhagen during a period of 12 years
114 (1848-1859). He measured outflow from the lakes and invented a system for reliable measurements
115 of the evaporation from a lake (Colding, 1860). A sheet metal box was placed on a float in the lake,
116 partly submersed such that the surface of the water in the box was approximately equal to the
117 surface of the lake. Hereby, the error due to different temperature in the lake and the box was
118 minimised. Moreover, the box was shielded to avoid errors due to wave splash. To include the
119 evaporation from wetlands in contact with the lake he supplemented the experiments by monitoring
120 evaporation from wetted short and long grass in which case he used weighting of the box, as the
121 water level in the box was difficult to assess. The monitoring efforts proved excellent for assessing
122 the amount of evaporation and its dependence on precipitation and seasonality. By monitoring the
123 lake outflow, he was able to obtain rather precise estimates of the lake inflow.

124 As the new water supply was strongly depending on groundwater utilisation, Colding was interested
125 also in getting an estimate of percolation to the groundwater. To that purpose, he used drain
126 experiments from which he obtained not just solid knowledge of the amount and seasonality in
127 percolation but also insight in the dependence of the soil type. This helped Colding to assess the



amount of water available for supply to Copenhagen. Colding maintained a lifelong interest in
evaporation, and it can be seen in his left papers that he was rather close to find an evaporation
formula close to Penman's (Penman, 1948).

**6. The free surface forms in conduits with constant flow**
During the experiments with flow in conduits, Colding became interested in investigating the
possible forms of the water table in steady, non-uniform flow in prismatic and cylindrical channels.
The starting point again was Eytelwein's formula for steady, uniform channel flow. He transformed
the coordinates to be parallel and perpendicular to the direction of the conduit an initiated a
complete mathematical-physical analysis. Depending of the slope of the conduit and the inflow and
outflow conditions, he could calculate the surface forms. For rectangular conduits, he identified and
described mathematically six different forms. In addition, the surface form due to damming of water
was considered.
Cylindrical conduits were analysed to the same degree of completeness. This resulted in
identification of 14 different water surface forms. Two examples of the surface forms in a circular
conduit, a concave and a convex, are shown in Fig 4. It is likely that Colding was the first to present
such a complete theory. The treatise (Colding, 1867), however, was written in Danish and therefore
not cited in international literature.

**7. Outflow of heat from pipes carrying hot water**
The reason for Colding to engage in this problem was his disagreement with a treatise by Dulong and
Petit who, based on a series of experiments, in 1818 had published a treatise casting doubt on
Newton's theory for heat transport. Colding's starting point was the heat equation by Poisson, which
he, by considering water loss from a pipe with flowing water, integrated and found in complete
agreement with Newton's theory. He also carried out several series of experiments measuring the
changing temperatures throughout a pipe with flowing. The experiment were outdoor and large-
scale, see Fig. 5, and performed in wintertime. Having found complete agreement with his
theoretical calculations, he continued with heat emission from the pipe after cessation of the flow.
In this case, the agreement with Newton was not immediately obtained, as the results rather
pointed to the correctness of Dulong and Petit. However, by taking into account the difference in
heat transport between flowing and stagnant water in the pipe he could, using advanced
mathematics, explain the apparent deviation (Colding, 1868).

**8. On the laws of currents in ordinary conduits and in the sea**
So far, Colding had successfully used Eytelwein's formula for the mean flow in prismatic and
cylindrical conduits with constant slope. He was, however, also interested in the velocity distribution
in a cross-section of the conduit. To that end, he used comprehensive measurements carried out by,
respectively, Boileau (1854), Darcy (1858) and Bazin (1865) to develop a complete theory based on
action and reaction throughout the cross section originating from the wall friction. Bolieau had
found that the maximum velocity in an open prismatic conduit might occur slightly below the
surface, which was in accordance during some experiments. Colding showed that this was
theoretical impossible without disturbances of the flow and found that even minor wind effects



could be the reason. Using advanced mathematics, Colding calculated the cross sectional velocity
distributions, which was convincingly verified by the French experiments (Colding, 1870). A couple of
examples are shown in Fig. 6.
In order to extend the theory to apply for currents in the sea with no walls to confine the current, he
collected all available information about the Golf Stream (i.a., U.S. Coast Survey, 1851; 1855; 1860;
Irminger, 1853; 1861; Maury, 1855; Forchhammer, 1859; Kohl, 1868). He did not agree with Maury
that the primary cause for the Golf Stream was a larger salinity in the Caribbean Sea than at higher
northern latitudes. Colding showed that the primary cause for the onset of the stream is the high
water level in the Caribbean due to the effect of the trade winds. He put forward a complete
physical/mathematical theory for the progress of the stream, including the total onset volume and
the volumes of the different branches the stream splits into when approaching the European
continent (Colding, 1870). He argued that the progress of the stream is dominated by the effect of
the earth's rotation, but also affected of many other elements like the return of the polar stream,
the net evaporation from the sea, and changing temperatures. Colding discussed in detail all these
factors. An overview of the Golf Stream from Colding's treatise is shown in Fig. 7. A strongly
abbreviated version of the treatise was later published in *Nature* (Colding, 1871b).

## 9. On the flow of air in the atmosphere

Colding had a strong interest in meteorological phenomena and applied the knowledge obtained
from the free currents in the sea to describe flow of air in the atmosphere. First, he showed that the
mathematical description of a rotating water whirl could be used to describe the movement of air in
a cyclone, see Fig. 9. The result were verified using observations of wind speed and air pressure
during the Antigua cyclone 2 August 1837 (Colding, 1871c). He then described the primary global
weather phenomena using the experience from the analysis of the Golf Stream. Unfortunately,
Colding did not include the influence of the rotation of the earth, the Coriolis' force (Coriolis 1835)
completely correct (not before the 20$^{th}$ century the Coriolis force began to be applied, first by
meteorologists). This was particularly critical for the large-scale wind systems, but not for the
cyclone theory. When he later realised the mistake, he initiated a revision of the free flow theory,
but died before it was completed.
Colding used the cyclone theory once more to assess the wind speeds around St. Thomas during the
cyclone that passed the island 21 August 1871. Air pressure observations from surrounding ships
were included, which made Colding able to obtain a detailed description of the track of the cyclone
and the winds speeds during the passage. The island was not damaged as much as during the 1837
cyclone, which corresponded well to Colding's assessment of the wind speeds and the storm track
(Colding, 1871d). The airflow theory was later published in German (Colding, 1875a).

## 10. On the laws for movement of water in soil

After establishment of a number of artesian groundwater wells with the first one drilled in 1851,
Colding closely followed the yield of the wells during the following years. He observed that the yield
varied seasonally with maximum in wintertime and between wet and dry years. By analysing the
yield as function of the piezometric head, he found the general law for movement of water in soil,
i.e. the proportionality between the head gradient and the velocity known as the Darcy equation. To
verify his findings, he established a series of experiments with water flowing through soil layers of





different types confirming the general law. Colding knew and used the pipe experiments by Darcy,
but was unaware of his groundwater work (Darcy, 1856). Thus, he independently came to the same
conclusion just a few years later. For a confined aquifer draining to open water, he found a parabolic
piezometric surface, see Fig. 9.
In parallel to the groundwater abstraction programme, Colding also established two sets of drain
experiments, one in sandy soil and one in clayey soil. As usual, Colding performed a profound
analytical treatment being, cf. Brutsaert (2005), the first to described the elliptic water table
between parallel drainpipes, see Fig. 10. The theoretical findings were accompanied by general
recommendations for establishment of drain system in agricultural fields (Colding, 1872).

**11. On the wind-induced currents in the sea**
The background for the last ground-breaking work of Colding was a partial flooding of the southern
Danish islands Lolland and Falster during a severe storm 12-14 November 1872, where 80 people
died and about 500 ships stranded during. Colding wanted to describe the cause of the flooding and
hypothesized that the wind forces' impact on the sea could fully explain the incident as the tidal
influence in the Baltic Sea is minimal. As a first necessary step, he developed a complete theory for
wind set-up in a wide channel including expressions for the set-up depending on the wind speed
relative to the original flow velocity (Colding, 1876).
Immediately after the storm, he had initiated a wide data collection, both nationally and
internationally, to get information on water levels, air pressure, and wind speed and direction. On
this basis, he developed synoptic weather maps including water levels for each six hours during the
storm. An example is shown in Fig. 11. For a number of sections in the Baltic Sea he subsequently
calculated the wind set-up according to his previous developed theory. This resulted a in a
remarkable match proving that in fact it was the wind that caused the flooding. Using the synoptic
maps, he was able to explain the development of the storm in detail. Finally, he added calculations
of the water flow through the Danish straits during the storm (Colding. 1881). Subsequent design of
dikes to prevent future flooding was based on Colding's theory.

**12. Final remarks**
Every second year between 1865 and 1883 Colding taught a course at the Polytechnic School on the
basic laws for discharge of sewage, water and gas supply, and heating and ventilation. His
handwritten lecture notes (Colding, 1875b) are still kept, see Fig. 12. In 1869, he was appointed as
Professor at the school. Two years before he was bestowed knight of the Order of Dannebrog, and in
1871 he received the honorary doctoral degree at the University of Edinburgh simultaneously with
Joule. Since 1856, he had been member of the Royal Danish Society of Sciences and Letters, and in
1875, he also joined the Royal Swedish Academy of Science. It is evident that Colding nationally
became highly valued in his lifetime both for his scientific achievements and for his endeavours for
the city of Copenhagen. However, even though he was a scientific frontrunner in many respects, he
seems nowadays almost forgotten. Maybe the above overview of his extremely diversified and
original research can lead to a renewed interest and appreciation?






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

Note: "Videnskabernes Selskab" refers to "The Royal Danish Society of Sciences and Letters"






**Figures**

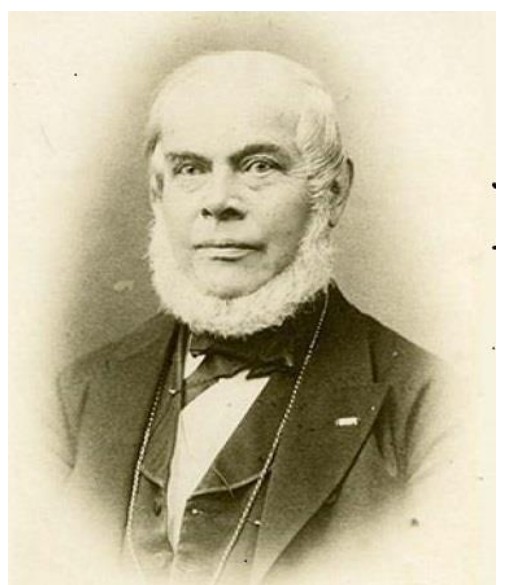


348                     Fig. 1. August Colding (Wikipedia).



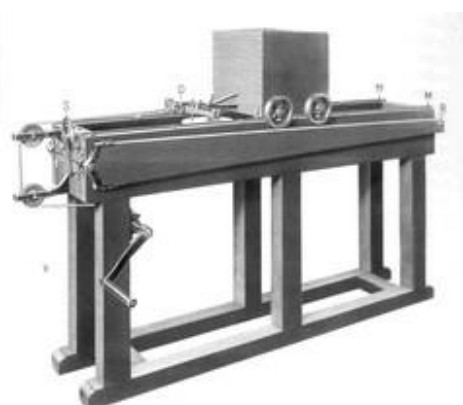


352       Fig. 2. The experimental set-up for the first law of thermodynamics (Danish Museum of Science and
353                                            Technology).






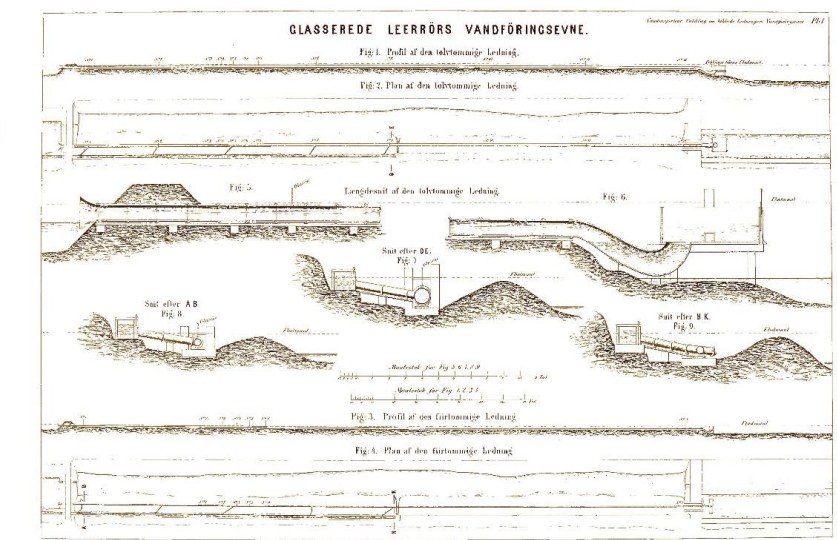

Fig. 3. Large-scale experiments with filled and partly filed conduits including inlets along the pipe (Colding, 1857).

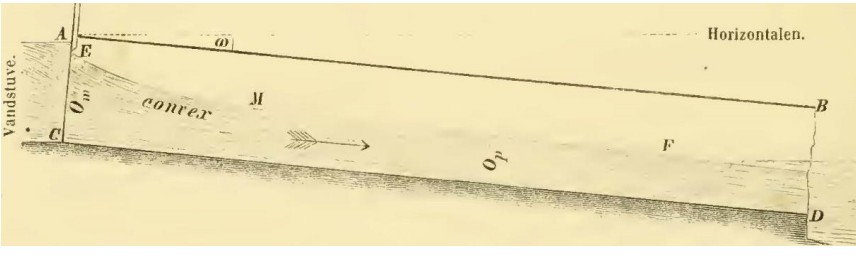

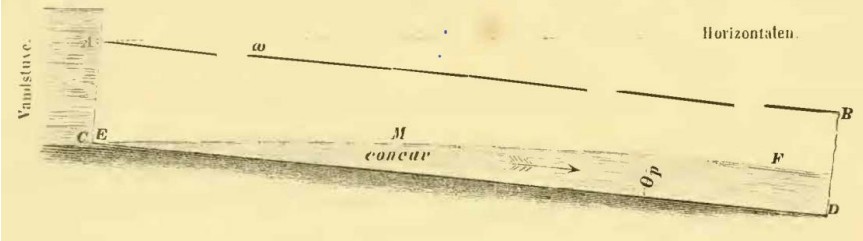

Fig. 4. Examples of surface forms in steady, non-uniform flow in circular conduits (Colding, 1867).







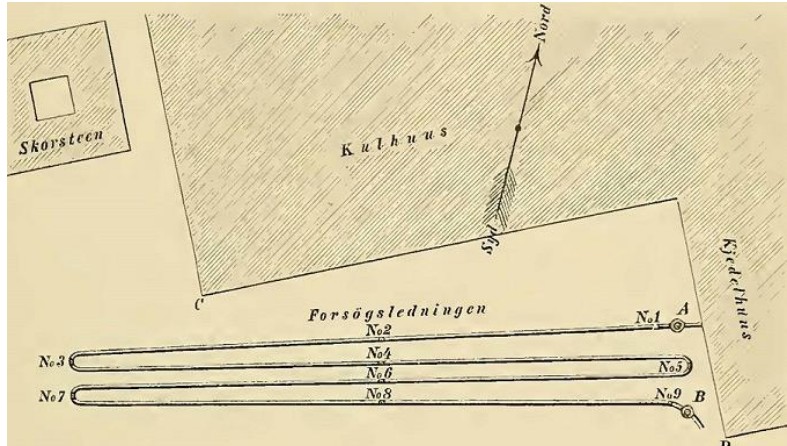


Fig. 5. The large-scale experimental set-up for measuring the heat loss from a pipe with hot water
(Colding, 1868).


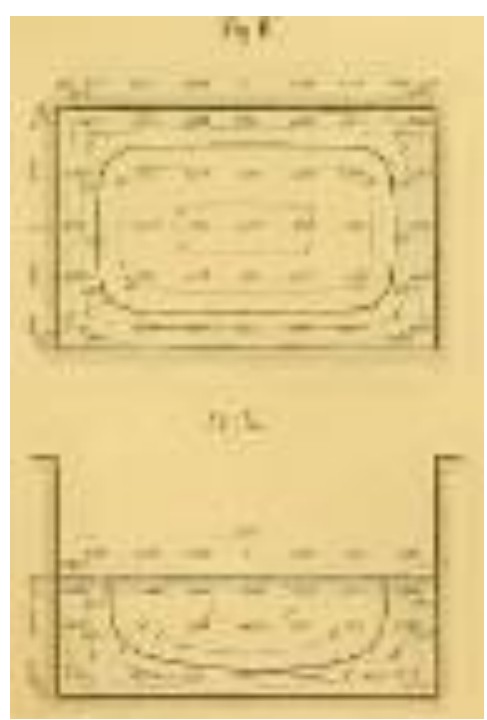


Fig. 6.  The cross-sectional velocity distribution in closed and open conduits (Colding, 1870).







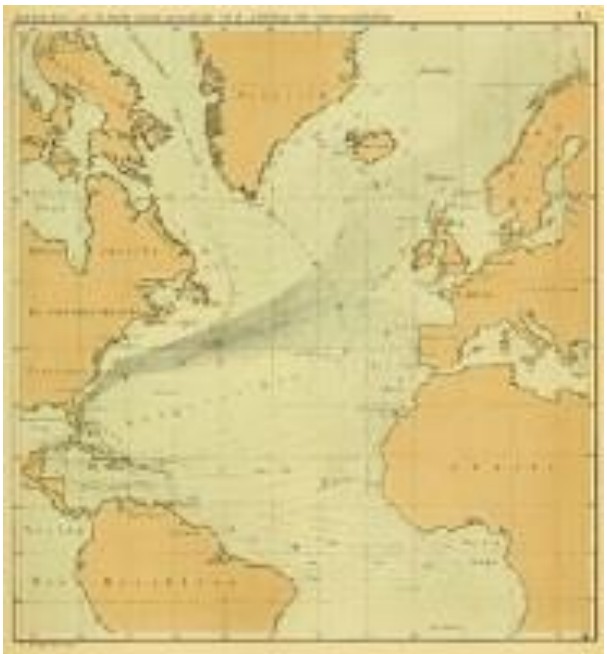


Fig. 7. The Golf Stream (Colding, 1870).


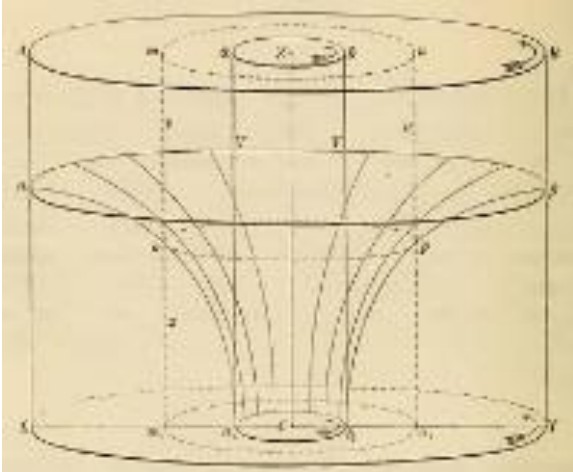


Fig. 8. The water whirl used as an analogue to a cyclone (Colding, 1871c).





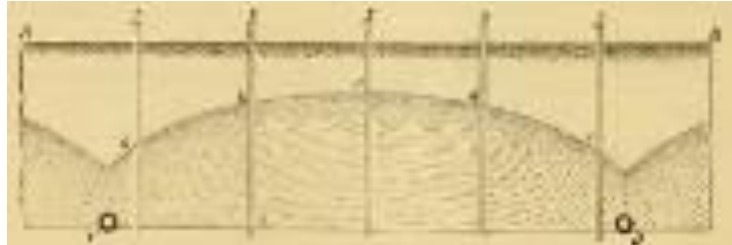


Fig. 9. Parabolic piezometric surface for a confined aquifer (Colding, 1872).



Fig. 10. Elliptic water table between drainpipes (Colding, 1872).

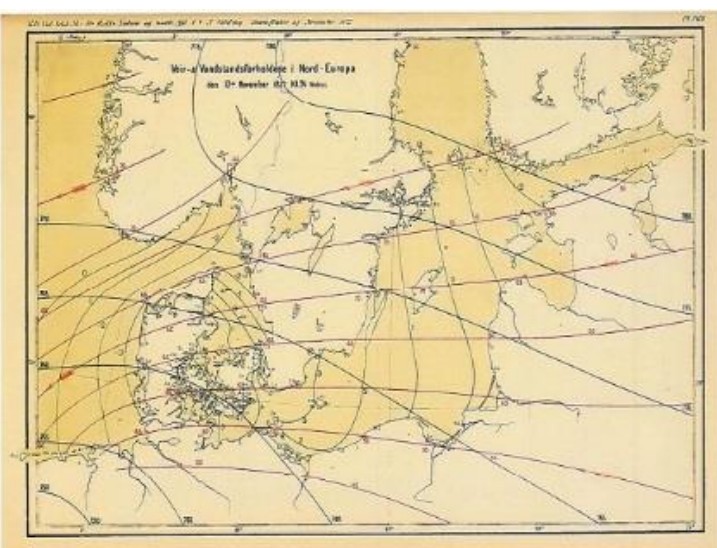


Fig. 11. Synoptic whether map from the 1872 storm in the Baltic Sea (Colding, 1881).





398        Fig. 12. Front page of Colding's handwritten 310 pages long lecture notes (Colding, 1875b).