# Peer review of "Hydrology and beyond: The scientific work of August Colding revisited"

_Hydrology and Earth System Sciences, 2020_

## Referee Comment (RC1) · Allan Rodhe (Referee) · 7 May 2020

The manuscript is an overview over the scientific work of a giant within environmental engineering, the Danish engineer and researcher August Colding (1815-88). His work is presented in ten short sections treating a wide range of topics within physics and earth sciences. The paper is well written and awakens the reader's interest in this researcher, who was strong in both theory and practical applications and who made several groundbreaking findings. It is an overview paper and within that format it has not been possible to go into details to give the reader a full understanding of the various experiments and theories. But the greatness of Colding is clearly shown and the reader can agree with the author's hope, formulated in the last sentence of the paper, "Maybe the above overview of his extremely diversified and original research can lead to a

renewed interest and appreciation". The paper is well worth publishing in Hess after taking the comments below into consideration.

Comments:

Several figures have poor quality and are difficult or impossible to interpret (Figure 3 and Figures 6 -11). Is it possible to scan the originals or better copies of the originals? The paper would be greatly improved with better figures. Explanatory captions would also be helpful.

Line 55-57 The author makes an important comment on the term "force". This could preferably be placed a little earlier, perhaps (slightly modified) as a footnote to line 44. With the present organization the reader gets confused by the use of "force" in the preceding paragraphs.

Line 121-123 The reader gets curious about data or other evidence behind the statement that "The monitoring efforts proved excellent for assessing the amount of evaporation and its dependence on precipitation and seasonality." Please describe Colding's evaluation.

Line 150-152 What type of "water loss from a pipe with flowing water" was considered? Please explain.

Line 215 and Figure 9 Did he really find a parabolic piezometric surface for a confined aquifer draining to open water? The shape of such surface is determined by the geometry of the confined aquifer (and the boundary conditions). The parabolic surface must be a special case. For an unconfined aquifer, on the other hand, the water table may have a parabolic surface (unconfined aquifer on a horizontal bottom, without recharge). Due to the poor quality of Figure 9 it is not possible to evaluate the experiment. Please improve the copy and explain what the figure shows.

There are some proof reading errors. See lines 153, 191, 219, 226, and 315 (1976 should be 1876).

---

## Referee Comment (RC2) · Anonymous Referee #2 · 2 Jul 2020

The pioneering work of August Colding has been summarized in this paper by Dan Rosbjerg.

The manuscript is generally well-written and it is interesting to read about Colding's independent work and it is evident that Colding pioneered the water sector in Denmark from both a theoretical and a utility application point of view in the latter part of the 19th century.

It is mentioned multiple times in the manuscript that Colding developed his theories simultaneously to other pioneering scientists such as Darcy, Joule et al, but was never recognized for his findings. The validity of these statements is difficult to prove and must been seen as postulates, and whether or not the experiments were purely independent can be questioned. I guess that the Code of Conduct for Research Integrity

was rather fussy at that time

The figures are of poor quality and give little insight into the actual experiments and development of theories. It is probably difficult to improve the quality of the original works, but a professional re-drawing might be an option if the reader should be able to make anything out of the figures.

Although the original works (which are well-cited) are in Danish, it could be interesting to include some citations and quotes from his original works. Maybe in Danish with a translation into English. This could contribute to improving the validity and scientific quality.

Whether the publication is suitable for the special issue on History of hydrology, I will leave for the editor to decide. The paper is interesting from a historical point of view, but it is mostly summarizing the works of Colding at a rather superficial and narrative level. The manuscript would fit well into a book on the history of hydrology. In my view, details of experiments and development of theories could be detailed more based on the original work in order for the manuscript to have any real scientific impact.

―――――――――――――――――

---

## Author Comment (AC1) · 19 Jul 2020

R#1: Several figures have poor quality and are difficult or impossible to interpret (Figure 3 and Figures 6 -11). Is it possible to scan the originals or better copies of the originals? The paper would be greatly improved with better figures. Explanatory captions would also be helpful.

Reply: Unfortunately, the quality of the figures in the pdfs of Colding's manuscripts I could get access to was rather poor. In my processing, the figure quality may have become even worse. I will try to improve the quality as much as possible. Moreover, I shall expand figure captions where appropriate.

R#1: Line 55-57 The author makes an important comment on the term "force". This

could preferably be placed a little earlier, perhaps (slightly modified) as a footnote to line 44. With the present organization the reader gets confused by the use of "force" in the preceding paragraphs.

Reply: As requested, I shall introduce the meaning of the term "force" earlier in the paper.

R#1: Line 121-123 The reader gets curious about data or other evidence behind the statement that "The monitoring efforts proved excellent for assessing the amount of evaporation and its dependence on precipitation and seasonality." Please describe Colding's evaluation.

Reply: I shall try to clarify and present Colding's evaluation in a better way.

R#1: Line 150-152 What type of "water loss from a pipe with flowing water" was considered? Please explain.

Reply: By mistake I wrote "water loss" instead of "heat loss". I shall correct.

R#1: Line 215 and Figure 9 Did he really find a parabolic piezometric surface for a confined aquifer draining to open water? The shape of such surface is determined by the geometry of the confined aquifer (and the boundary conditions). The parabolic surface must be a special case. For an unconfined aquifer, on the other hand, the water table may have a parabolic surface (unconfined aquifer on a horizontal bottom, without recharge). Due to the poor quality of Figure 9 it is not possible to evaluate the experiment. Please improve the copy and explain what the figure shows.

Reply: Approximately, he found a parabolic piezometric surface, resembling that steady, uniform recharge to a horizontal confined layer of constant thickness with fixed boundary conditions leads to a parabolic solution.

R#1: There are some proof reading errors. See lines 153, 191, 219, 226, and 315 (1976 should be 1876).

Reply: The errors will be corrected.

Thanks for the comments, they have been very helpful.

---

## Author Comment (AC2) · 19 Jul 2020

R#2: It is mentioned multiple times in the manuscript that Colding developed his theories simultaneously to other pioneering scientists such as Darcy, Joule et al, but was never recognized for his findings. The validity of these statements is difficult to prove and must been seen as postulates, and whether or not the experiments were purely independent can be questioned. I guess that the Code of Conduct for Research Integrity was rather fussy at that time.

Reply: The message was not that Colding was not recognized at all, but that it has been far from the level he might deserve. He cannot be found in international hydrological literature except for the short remark in the book of Brutseart referring to second hand

information without a proper reference to Colding's paper. Joule published on the first law of thermodynamics in 1843, the same year in which Colding submitted his treatise. A year before, Robert Mayer had published on the subject. I shall add references to Mayer and Joule. All three claimed priority to the finding. The real break-through for the first law of thermodynamics, however, came first after1847, where Herman Helmholz published the book "On the conservation of force". He referred to Joule, but later he also recognised the pioneering works for Mayer and Golding. Since the first law of thermodynamics is not the focus of the paper, I have chosen not to elaborate on the subject. In the work Colding (1871b) there are several references to Darcy's experiments on pipe flow, but no one in the work Colding (1872) on groundwater. I find this a strong indication for Colding's unawareness of Darcy's groundwater paper.

R#2: The figures are of poor quality and give little insight into the actual experiments and development of theories. It is probably difficult to improve the quality of the original works, but a professional re-drawing might be an option if the reader should be able to make anything out of the figures.

Reply: Unfortunately, the quality of the figures in the pdfs of Colding's manuscripts I could get access to was rather poor. In my processing, the figure quality may have become even worse. I shall try to improve the quality as much as possible. Regrettably, a professional redrawing is not an option.

R#2: Although the original works (which are well-cited) are in Danish, it could be interesting to include some citations and quotes from his original works. Maybe in Danish with a translation into English. This could contribute to improving the validity and scientific quality.

Reply: The writing style of Cloding is rather wordy. I doubt that it will fit well into the paper to include direct citations from Colding's work and translations of these into English. It would inappropriately expand the size of the paper.

R#2: Whether the publication is suitable for the special issue on History of hydrology, I

will leave for the editor to decide. The paper is interesting from a historical point of view, but it is mostly summarizing the works of Colding at a rather superficial and narrative level. The manuscript would fit well into a book on the history of hydrology. In my view, details of experiments and development of theories could be detailed more based on the original work in order for the manuscript to have any real scientific impact.

Reply: My intention has been to present Colding's work in a relatively short form with primary focus on the hydrologically related subjects. Without changing this intention, I may elaborate slightly more on some of the subjects.

Thanks for the comments, they have been very helpful.

---

## Author Response (AR1)

**Response to R#1 Allan Rodhe**

R#1: Several figures have poor quality and are difficult or impossible to interpret (Figure 3 and Figures 6 - 11). Is it possible to scan the originals or better copies of the originals? The paper would be greatly improved with better figures. Explanatory captions would also be helpful.

Reply: The figure quality is improved as much as possible, and figure captions are expanded where appropriate.

R#1: Line 55-57 The author makes an important comment on the term "force". This could preferably be placed a little earlier, perhaps (slightly modified) as a footnote to line 44. With the present organization the reader gets confused by the use of "force" in the preceding paragraphs.

Reply: The term "force" is introduced earlier in the paper.

R#1: Line 121-123 The reader gets curious about data or other evidence behind the statement that "The monitoring efforts proved excellent for assessing the amount of evaporation and its dependence on precipitation and seasonality." Please describe Colding's evaluation.

Reply: Colding's evaluation is clarified and presented in a better way.

R#1: Line 150-152 What type of "water loss from a pipe with flowing water" was considered? Please explain.

Reply: "water loss" is corrected to "heat loss".

R#1: Line 215 and Figure 9 Did he really find a parabolic piezometric surface for a confined aquifer draining to open water? The shape of such surface is determined by the geometry of the confined aquifer (and the boundary conditions). The parabolic surface must be a special case. For an unconfined aquifer, on the other hand, the water table may have a parabolic surface (unconfined aquifer on a horizontal bottom, without recharge). Due to the poor quality of Figure 9 it is not possible to evaluate the experiment. Please improve the copy and explain what the figure shows.

Reply: The parabolic piezometric surface is maintained, resembling steady, uniform recharge to a horizontal confined layer of constant thickness with fixed boundary conditions. The figure is explained in a better way.

R#1: There are some proof reading errors. See lines 153, 191, 219, 226, and 315 (1976 should be 1876).

Reply: Corrected.

**Response to R#2 anonymous**

R#2: It is mentioned multiple times in the manuscript that Colding developed his theories simultaneously to other pioneering scientists such as Darcy, Joule et al, but was never recognized for his findings. The validity of these statements is difficult to prove and must been seen as postulates, and whether or not the experiments were purely independent can be questioned. I guess that the Code of Conduct for Research Integrity was rather fussy at that time.

Reply: The text has been expanded to account for the referee's concern.

R#2: The figures are of poor quality and give little insight into the actual experiments and development of theories. It is probably difficult to improve the quality of the original works, but a professional re-drawing might be an option if the reader should be able to make anything out of the figures.

Reply: The figure quality is improved as much as possible.

R#2: Although the original works (which are well-cited) are in Danish, it could be interesting to include some citations and quotes from his original works. Maybe in Danish with a translation into English. This could contribute to improving the validity and scientific quality.

Reply: A few citations in Danish with English translation are included in the text.

R#2: Whether the publication is suitable for the special issue on History of hydrology, I will leave for the editor to decide. The paper is interesting from a historical point of view, but it is mostly summarizing the works of Colding at a rather superficial and narrative level. The manuscript would fit well into a book on the history of hydrology. In my view, details of experiments and development of theories could be detailed more based on the original work in order for the manuscript to have any real scientific impact.

Reply: Selected subjects have been elaborated.

**Response to Editor Keith Beven**

Editor: I hope you will be able to prepare a revision of your paper, taking account of the referees comments and improving the figures as far as possible. There some aspects in your responses to the referees comments that could usefully include in the paper (e.g. in relation to Darcy). I wonder if it might be interesting to include one example of Colding's writing if you can find something pertinent to quote – even if wordy it would give more of an impression of his style.

Reply: In the revision, I considered all the referee comments including improvement of the figures, enlargement of the section on groundwater and inclusion of a couple of examples of Colding's writing.

[revised manuscript text omitted]